# The Multifaceted Role of Serotonin in Intestinal Homeostasis

**DOI:** 10.3390/ijms22179487

**Published:** 2021-08-31

**Authors:** Nienke Koopman, Drosos Katsavelis, Anne S. ten Hove, Stanley Brul, Wouter J. de Jonge, Jurgen Seppen

**Affiliations:** 1Swammerdam Institute for Life Sciences (SILS), University of Amsterdam, 1098XH Amsterdam, The Netherlands; n.koopman@amsterdamumc.nl (N.K.); droskats@yahoo.com (D.K.); s.brul@uva.nl (S.B.); 2Tytgat Institute for Liver and Intestinal Research, Amsterdam University Medical Centers, Location AMC, 1105BK Amsterdam, The Netherlands; a.s.tenhove@amsterdamumc.nl (A.S.t.H.); w.j.dejonge@amsterdamumc.nl (W.J.d.J.)

**Keywords:** inflammatory bowel disease, intestine, tryptophan, microbiome

## Abstract

The monoamine serotonin, 5-hydroxytryptamine (5-HT), is a remarkable molecule with conserved production in prokaryotes and eukaryotes and a wide range of functions. In the gastrointestinal tract, enterochromaffin cells are the most important source for 5-HT production. Some intestinal bacterial species are also able to produce 5-HT. Besides its role as a neurotransmitter, 5-HT acts on immune cells to regulate their activation. Several lines of evidence indicate that intestinal 5-HT signaling is altered in patients with inflammatory bowel disease. In this review, we discuss the current knowledge on the production, secretion, and signaling of 5-HT in the intestine. We present an inventory of intestinal immune and epithelial cells that respond to 5-HT and describe the effects of these signaling processes on intestinal homeostasis. Further, we detail the mechanisms by which 5-HT could affect inflammatory bowel disease course and describe the effects of interventions that target intestinal 5-HT signaling.

## 1. Introduction

The monoamine serotonin, or 5-hydroxytryptamine (5-HT), derived from tryptophan, is a neurotransmitter and signaling molecule. In this review, we discuss the current knowledge on the action of 5-HT in intestinal homeostasis with a focus on its non-neuronal effects. This review provides a comprehensive and up-to-date overview of the role of 5-HT in intestinal homeostasis with an emphasis on molecular pathways in inflammatory bowel disease (IBD). We expand on existing literature [1,2,3,4] by giving a thorough outline of non-neuronal cell types that can respond to 5-HT signaling. In addition, we describe the importance of the microbiome in the production of 5-HT and in the modulation of 5-HT signaling. Finally, the effects of therapeutic interventions targeting 5-HT signaling in the gut are reviewed.

## 2. 5-HT in Intestinal Homeostasis

### 2.1. 5-HT Metabolism in the Gut

5-HT is mainly known for its action on the central nervous system and its key role in the regulation of mood and behavior. However, only approximately 5% of the total 5-HT in the human body is produced by serotonergic neurons in the central nervous system. The other 95% is produced in the intestine, where it also plays a major role in the regulation of the enteric nervous system (ENS), immune responses, and epithelial integrity [5,6].

#### 2.1.1. Endogenous 5-HT Synthesis

In peripheral tissues, 5-HT biosynthesis takes place in a variety of cells, such as adipocytes [7], pancreatic β cells [8], and osteoclasts [9]. In addition, immune cells like monocytes [10,11], mast cells [12], macrophages [10,13], dendritic cells (DCs) [14], B cells [15], and T cells [13,16] express tryptophan hydroxylase (TpH) enabling 5-HT production. Intriguingly, in the gastrointestinal tract, 90%, the greatest level of enteral 5-HT, is synthesized by enterochromaffin (EC) cells. The myenteric plexus of the ENS is responsible for the synthesis of the remaining 5-HT production in the gut (Table 1) [17].

The essential amino acid tryptophan is the precursor for the biosynthesis of 5-HT [5]. This metabolic pathway is biphasic. First, the rate-limiting enzyme TpH catalyzes the conversion of L-tryptophan to 5-hydroxytryptophan (5-HTP) [18]. Second, a rapid conversion takes place; of 5-HTP to 5-HT by the enzyme aromatic L-amino acid decarboxylase [18]. Of TpH, two isoforms have been identified: TpH1, which is mainly expressed by EC cells and is responsible for most peripheral 5-HT production, and TpH2, which is expressed by enteric neurons and serotonergic neurons in the brain [19,20]. TpH1-derived 5-HT is involved in intestinal inflammation via the activation of immune responses, whereas 5-HT derived from TpH2 in the ENS regulates gut motility and intestinal neurogenesis [19].

#### 2.1.2. Release and Reuptake of 5-HT

EC cells are able to sense signals in the gut lumen, such as pH, nutrient changes, toxins, or neuromodulatory agents, and respond to mechanical stimuli, such as intestinal muscle contractions causing the release of 5-HT. For this, firstly, 5-HT is sequestered into granules by the vesicular monoamine transporter 1 (VMAT1) [9,26]. These granules are then transferred to the membrane, where 5-HT is released with the help of the serotonin reuptake receptor (SERT) on the basal side in the lamina propria or in the intestinal lumen on the apical side of the membrane [18,24,25]. Released 5-HT can be taken up via SERT by mucosal enterocytes, where it can be metabolized [23]. Here, the majority of 5-HT is degraded by monoamine oxidase (MAO), an enzyme located in the mitochondria that catalyze the deamination of many biogenic amines [27]. The final products of this metabolic pathway are either 5-hydoxyindoleacetic acid or, in insignificant amounts, 5-hydroxytryptophol, which are excreted via urine [28]. In enterocytes, 5-HT can also be metabolized by 5′-diphospho-glucuronosyltransferase to its inactive metabolite 5-hydroxytryptamine (5-HTO) glucuronide [29].

Most of the 5-HT released from EC cells enters the blood circulation via capillary beds in the submucosal part of the intestinal wall. The larger proportion of 5-HT is taken up in blood platelets via SERT, where it is packaged again into granules by VMAT1 or can be degraded by a variety of intracellular enzymes, including MAO [21,30,31]. Due to the lack of TpH, platelets are not able to synthesize 5-HT and merely act as transporter [22]. A small percentage of 5-HT stays soluble in plasma and binds to 5-HT-specific receptors in peripheral tissues [18].

Neuronal 5-HT produced in the ENS is released by presynaptic neurons and activates postsynaptic 5-HT receptors (HTRs or 5-HTRs) [3]. Subsequently, it can be transported back into the presynaptic neurons by SERT, where it is deactivated.

#### 2.1.3. Microbial Synthesis of 5-HT

Only a small part of by the diet available tryptophan is converted to 5-HT by the host (1–2%). The vast majority enters the kynurenine pathway, where it is converted by indolamine-2,3-dioxygenase (IDO) to different bioactive substances, such as kynurenic, picolinic, and quinolinic acid [32,33]. These other tryptophan derivatives interact with the aryl hydrocarbon receptor, a cytosolic ligand-activated transcription factor encoding genes that modulate immune responses, maintain the epithelial barrier function, and prevent intestinal inflammation [34]. An important fraction of the dietary tryptophan (4–6%) is also metabolized by the gut microbiota [32]. In addition, minimal proportions of dietary tryptophan are used by mammals for protein synthesis [35], and finally, approximately 0.5% is excreted in urine [36].

Although tryptophan metabolism by gut microbiota is limited, substantially reduced, serum concentrations of 5-HT and increased concentrations of tryptophan were demonstrated in germ-free mice when compared with conventionally raised mice [37,38,39,40,41,42]. These levels can be restored by colonizing germ-free mice with gut microbiota from feces of conventionally raised mice or human feces [38,40,42]. Similar results were observed in conventionally raised mice or rats treated with antibiotics [37,43,44]. This is as bacteria are shown to compete with the host for dietary tryptophan by taking up tryptophan and converting this into derivatives such as indole and skatole [32,33]. In addition, commensal bacteria in the colon, such as *Clostridium sporogenes* and *Ruminococcus gnavus*, which are associated with IBD, and are able to take up tryptophan and degrade it to tryptamine through decarboxylation [45,46,47,48]. Some bacterial species express tryptophan synthase allowing for tryptophan synthesis contributing to the tryptophan pool [49].

Interestingly, certain bacterial species have shown in vitro to be able to convert tryptophan to 5-HT (Table 2) [50,51,52,53]. Yet, it still remains unknown what the physiological function of 5-HT is in bacteria, whether microbial 5-HT production also takes place in vivo in mammals, and if so, whether the produced concentrations have any biological relevance for the host. In short, affecting the tryptophan availability is not the only way in which the microbiome influences 5-HT metabolism, as 5-HT levels can also be altered directly. Figure 1 gives an overview of the production of 5-HT in the intestine.

IBDs, including Crohn’s disease (CD) and ulcerative colitis (UC), are characterized by chronic intestinal inflammation [54]. Currently, it affects up to 0.5% of the world population, and a higher prevalence in areas adopting a Western lifestyle is observed [55]. Patients with IBD show an altered gut microbiome composition, mainly characterized by a decrease in microbial diversity, including a decrease in Firmicutes with a depletion in *Clostridium* cluster IV and XIV and an increase in *Enterobacteriaceae* species [56]. When comparing bacterial families altered in IBD to a large co-abundance network, it can be concluded that the Firmicutes phylum members of the *Streptococcaceae* family were increased in patients with CD and UC while the abundance level of the *Lactobacillaceae* family was found increased in patients with CD but reduced in patients with UC [57]. Bacteria from the *Enterobacteriaceae* family of the Proteobacteria phylum were more abundant in patients with CD and less so in patients with UC [57]. These bacteria are all capable of producing 5-HT, and hence, these data indicate that microbial 5-HT metabolism is potentially altered in patients with IBD.

#### 2.1.4. Modulation of Host 5-HT Synthesis by Gut Microbiota

In addition to the direct effects on tryptophan availability, animal models showed that the gut microbiota also indirectly affect the tryptophan availability by activating host genes encoding for enzymes involved in tryptophan catabolism via the kynurenine pathway [58,59]. For example, certain bacterial strains, such as *Bifidobacterium infantis*, were shown to increase plasma tryptophan levels when administered to rats by reducing IDO activity in the kynurenine pathway. As a result, more available tryptophan can be used from the host for 5-HT synthesis [59].

Microbial metabolites are also shown to directly affect the 5-HT production by the host. In the colons of germ-free mice, the expression levels of TpH1 are shown to be reduced, while the expression levels of the enzymes regulating 5-HT storage, release, and degradation remain were unchanged. Additionally, many studies have demonstrated that gut microbiota, particularly spore-forming microbes, can promote 5-HT production from EC cells via TpH1 stimulation [37]. The most well-studied metabolites affecting 5-HT synthesis are short-chain fatty acids (SCFAs) and secondary bile acids, especially deoxycholate. SCFAs, such as acetate, propionate, and butyrate, result from the anaerobic fermentation of dietary fibers in the intestine and are produced by bacteria belonging to Firmicutes and Bacteroidetes, the two most dominant phyla in the human gut microbiome [60]. Microbial and dietary SCFAs stimulate the free fatty acid receptors in EC cells, which increases TpH1, and therefore, 5-HT production [60]. Bile acids can activate the G-protein coupled bile acid receptor TGR5, expressed by EC cells and intrinsic primary afferent neurons, to stimulate the secretion of 5-HT [61]. Recent experiments revealed that other microbial metabolites like α-tocopherol, cholate, tyramine, and p-aminobenzoate also increase 5-HT secretion from EC cells, while other studies showed that the same effect is caused by bacterial toxins, such as the cholera toxin and *Escherichia coli* lipopolysaccharide (LPS) [37,62]. Microbial-produced tryptamine can also induce 5-HT production from myenteric neurons [63].

Microbes are also able to increase 5-HT availability in other ways. Germ-free mice colonized by pathogen-free microbes show a small but significant proportion of 5-HT, which is derived from the deconjugation of 5-HTO glucuronide in the liver and ends up in the gut lumen through the bile duct [64]. This process requires the bacterial enzyme β-glucuronidase [64]. Secondly, host SERT expression and functioning seem to be altered by the presence of gut microbiota. SERT-mediated secretion was shown to be upregulated in germ-free mice, and pathogenic *Escherichia coli* was shown to inhibit SERT expression and function [37,65].

#### 2.1.5. Host 5-HT Metabolism Affects Microbiota Composition and Functioning

In vitro studies suggest that 5-HT has an effect on bacterial growth, as increased growth and promoted cell aggregation has been observed in *E. coli* and *Rhodospirillum rubrum* cultures when 2 × 10^−5^ M 5-HT was administered. However, opposite results were observed when stimulating in higher concentrations [66]. In line, 5-HT availability can influence the gut microbiota composition as TpH1^+/−^ and TpH1^−/−^ mice, exhibiting different amounts of intestinal 5-HT, were shown to have differences in gut microbiota compositions. 5-HT was also shown to directly inhibit the growth of beneficial bacteria at the species level, as well as SCFA production through inhibition of obligate anaerobe growth in a concentration-dependent manner [62].

Germ-free mice colonized by gut microbiota from TpH1^+/−^ mice after dextran sulfate sodium (DSS) treatment inducing colitis contained less *Akkermansia muciniphila* and demonstrated more severe colitis than germ-free mice colonized with TpH1^−/−^ microbiota. This bacterium uses mucin as an energy source to produce SCFAs, forcing goblet cells to produce more mucus. Hence, it contributes to the strengthening of the gut barrier and the modulation of immune responses [67]. Moreover, 5-HT can reduce peroxisome proliferator-activated receptor (PPAR)-γ expression and thereby the production of the antimicrobial peptide β-defensin [68]. Interestingly, *Bacteroides thetaiotaomicron* and *Enterococcus faecalis* have been shown to activate intestinal epithelial PPAR-γ [69,70]. In other experiments, 5-HT was shown to inhibit the activation of PPAR-γ by suppressing the growth of these microbes [62]. Another study showed that increased 5-HT availability in mice via oral administration or by SERT deletion increased the amount of spore-forming bacteria [71].

Further evidence for the direct effect of 5-HT availability on gut microbes is given by SERT^−/−^ mice which have altered bacterial composition in both fecal and cecal samples. More specifically, these mice show higher levels of Bacilli, including the genera *Lactobacillus*, *Streptococcus*, *Enterococcus*, and *Listeria*, but significantly lower amounts of *Bifidobacterium* species and *Akkermansia muciniphilia* [72]. In addition, *Turicibacter sanguinis*, a common gut microbe found in the gut lumen, expresses a membrane protein, named CUW_0748, that is homologous to SERT and is able to take up 5-HT from the gut environment [71]. In cases of excessive 5-HT availability, like in inflammatory circumstances [73,74,75,76], *T. sanguinis* colonization is increased, and the host’s lipid metabolism is altered [71]. It is hypothesized that SCFA producing bacteria, such as *T. sanguinis*, have co-evolved to stimulate 5-HT synthesis, which they subsequently use to successfully colonize the intestine. Interestingly, the *Turicibacter* genus is found to be increased in samples from IBD patients [77].

### 2.2. Signaling of 5-HT

#### 2.2.1. HTRs in the Immune System

Monoamines, including 5-HT, produced in the gastrointestinal ENS, can influence immune responses through activation of HTRs found on immune cells [6,78]. As demonstrated in Table 3, almost every receptor or receptor subtype is distributed over more than one type of immune cell, creating a diverse and complex network of HTRs in the immune system with various functions. At least 14 mammalian HTRs are divided into distinct classes (HTR1–7) based on their functional, structural, and transductional characteristics [79]. This heterogeneity is a result of cellular mechanisms, such as RNA editing, alternative splicing, homo- and heterodimerization, and polymorphic variants [79]. All classes, except for HTR3, contain G-protein coupled receptors (GPCRs) activating an intracellular second messenger cascade [80]. Adenylyl cyclase is negatively coupled to HTR1 and HTR5 receptors which means that their activation downregulates cAMP [80]. Contrarily, activation of HTR4, HTR6, and HTR7 receptors increase cAMP levels [80]. The HTR2 class is associated with the upregulation of inositol triphosphate and diacylglycerol pathways that lead to a rise in intracellular Ca^2+^ [79]. The HTR3 is a Cys-loop ligand-gated ion channel, and its activation causes a rapid depolarization of the plasma membrane [81].

#### 2.2.2. Serotonylation

Serotonylation is a receptor-independent mechanism of 5-HT signaling [98]; it is the creation of glutamyl-amine bonds between 5-HT and small GTPases, such as Rho and Rab4, catalyzed by the enzyme transglutaminase in platelets. This interaction prevents GTP hydrolysis and results in the exocytosis of α-granules from platelets. Serotonylation could be involved as pathophysiological effect of abnormal 5-HT levels in IBD, as these GTPases can also be found in lymphocytes and other immune cells [99].

### 2.3. Role of 5-HT in Inflammation

During intestinal inflammation, the number of EC cells is often increased, which may result in higher levels of released 5-HT [29]. In normal conditions, the majority of 5-HT is transported through SERT to enterocytes for degradation or in blood platelets for storage [21,23,31], but in intestinal inflammation, excessive 5-HT activates local immune cells via HTRs. Immune cells like T cells, dendritic cells (DCs), and macrophages are able to trigger the activation of pro-inflammatory pathways and the secretion of pro-inflammatory cytokines. On top of that, released 5-HT acts as a chemoattractant recruiting leukocytes to the inflammatory site [11]. These processes will be described in the sections below. In addition, when stimulated by microbes, soluble factors such as platelet activation factors or IgE-containing structures, platelets secrete 5-HT, which subsequentially activates other platelets via HTR2A and HTR3, resulting in intracellular Ca^2+^ release and stabilization of platelet activation [11,93]. This also happens upon endothelial interactions. Table 4 summarizes the effect of 5-HT signaling on different cell types in the intestine.

#### 2.3.1. 5-HT Signaling on Monocytes

5-HT affects monocyte signaling in at least three pathways. First of all, the adhesion of monocytes to colonic epithelial cells, a process important in intestinal inflammation, is regulated by 5-HT via NADPH oxidase 2-derived reactive oxygen species [105]. This process results in the upregulated secretion of pro-inflammatory cytokines IL-6, IL-8, and the monocyte chemoattractant protein-1 [105]. Secondly, 5-HT binding on HTR4 reduces TNF-α levels in a concentration-dependent manner [88]. Thirdly, co-stimulation of monocytes by LPS and 5-HT increases IL-1β, IL-6, and IL-8 secretion via the activation of HTR3, HTR4, and HTR7 and enhances the production of IL-12p40 when HTR7 is activated. IL-12p40 can subsequently act as a chemoattractant for macrophages and bacterially activated DCs [88].

#### 2.3.2. 5-HT Signaling on Macrophages

HTRs are expressed in macrophages, predicting multiple functions [82]. Macrophages harvested from the peritoneal cavity of TpH1^+/+^ mice showed increased levels of IL-1β, and IL-6 in the presence of 5-HT, regardless of LPS stimulation [104]. The production of these pro-inflammatory cytokines was reduced after an NF-κB inhibitor was added to the cells. In another study, 5-HT was shown to increase the phagocytosis activity of peritoneal macrophages through HT1A receptors. This phagocytosis activity could be inhibited by an NF-κB inhibitor in a dose-dependent manner, in the presence but not in the absence of 5-HT [86].

Other studies showed that 5-HT is also able to activate macrophages through HTR2C [87], HTR2B, and HTR7 [10]. Interestingly, activation of the latter two was shown to induce an anti-inflammatory polarization of macrophages [10].

#### 2.3.3. 5-HT Signaling on DCs

DCs can be stimulated by 5-HT to produce pro-inflammatory cytokines and induce migration [83]. The maturity status of DCs is suggested to determine the effect of 5-HT. For instance, activated HTR1 and HTR2 cause the migration of immature DCs [101], while HTR7 is necessary for the migration and morphology of mature DCs and the chemokine receptor CCR7 and the Rho-GTPase Cdc42 are also involved [102]. In addition, it was shown that activation of HTR3, HTR4, and HTR7 induces expression of IL-6 [101] and that activation of HTR4 and HTR7 upregulates IL-1β and IL-8 in mature human DCs in an NF-κΒ dependent manner. These interleukins (ILs) are known to induce migration and neutrophil recruitment [83].

#### 2.3.4. 5-HT Signaling on T Cells

5-HT plays a key role in the interaction between DCs and T cells. When activated, these cells have the ability to synthesize 5-HT through TpH1 that is released and stored in DCs via SERT [14]. Interaction between the two cell types induces the release of Ca^2+^ from the DCs and causes the secretion of 5-HT from the lysosomal-associated membrane protein 1 vesicles, subsequently promoting T cell proliferation and differentiation of naive T cells [14].

5-HT binding of HTR7 of activated and naive T cells promotes their proliferation via the phosphorylation of extracellular signal-related kinase-1, and -2, and nuclear factor of kappa light polypeptide gene enhancer in B cell inhibitor alpha (IκBα) in naive T cells [89]. This results in enhanced expression of HTR1B, inducing proliferation of T helper (Th) cells [90] and HTR2A, which is responsible for increased production of IFN-γ and IL-2 from antigen-specific Th1 and cytotoxic T lymphocytes [91]. Th17 cells and T regulatory cells (Treg) are two lines of T cells that act in opposite ways and are involved in autoimmune inflammations [6]. While Th17 cells promote inflammation, Treg cells are supposed to be negative regulators of immune responses, thus suppressing autoimmunity [6]. This balance is also shown to be necessary for the maintenance of the gut microbiota composition allowing the beneficial microbes to colonize the gut, a process that is possibly regulated by 5-HT as described above [110]. Moreover, it was shown that 5-HT binding to the HTR1A on activated human T cells modulates intracellular levels of cAMP [92], while T cells are able to produce SERT [16].

#### 2.3.5. 5-HT Signaling on Granulocytes

Neutrophils can be indirectly recruited to inflammation sites by ILs produced by other immune cells, such as DCs or macrophages, after their activation by 5-HT [83]. In addition, they can be stimulated by platelet-derived chemotactic factors, such as platelet-activating factor and histamine, or through the expression and platelet-derived 5-HT [106]. Conversely, 5-HT and IL-3 synergistic action prevents the secretion of histamine, IL-4, and IL-6 from murine basophils and block the production of IL-13 and IL-4 from human peripheral blood basophils [99].

In mast cells, 5-HT seems unable to cause cytokine production or degranulation. However, it was shown that activation of HTR1A induces their migration and adhesion to fibronectin, an important step in the extravasation of mast cells to the inflamed tissue [85].

Little is known about the 5-HT function on eosinophils, but it is shown that 5-HT triggers the recruitment and adhesion of eosinophils to vascular cell adhesion molecule-1 (VCAM-1) through the activation of HTR2A [84].

#### 2.3.6. 5-HT Signaling on Natural Killer Cells and B Cells

5-HT affects natural killer (NK) cell immune properties by repressing their interaction with monocytes, resulting in increased NK cell cytotoxicity [107]. It was suggested that HTR1A might play a role in the regulation of a cell-to-cell mediated interaction between monocytes and NK cells [107]. In B cells, which are capable of uptake and releasing of 5-HT via SERT production [15], 5-HT is involved by promoting proliferation through the activation of HTR1A [100].

#### 2.3.7. 5-HT Signaling in Epithelial Cells

In the small intestine of HTR2A^−/−^ mice, the density of Paneth cells and the size of enterocytes were shown to be significantly reduced in comparison to wild-type littermates [94]. This could indicate that these receptors are involved in the development and maintenance of end-stage epithelial cells in intestinal crypts [94]. Whether these receptors also play a role in maintaining the barrier function still needs to be investigated.

Commensal bacterium *Bifidobacterium dentium* was shown to be able to induce 5-HT production by EC cells which in turn activated HTR4 on goblet cells and thereby promoted Muc2 and Trefoil factor 3 (TFF3) release [96]. TFF3 activates its receptor CXCR4 to promote actin cytoskeleton rearrangement and epithelial repair [96]. HTR4 on goblet cells was also highlighted in another study in which the activation of these receptors promoted goblet cell degranulation [95]. 5-HT4R stimulation on EC cells, goblet cells, and enterocytes could promote colonic transit [95].

### 2.4. 5-HT Signaling in Inflammation, Gut Motility, and Wound Healing

Alterations in 5-HT metabolism have been well described in patients with IBD. Patients with CD show upregulated expression of TpH1, 5-HTR3, 5-HTR4, and 5-HTR7 and downregulated expression of SERT in colonic tissues [111,112,113]. Evidence from experiments with samples from patients with IBD or animal models with experimentally-induced colitis have indicated clear bi-directional interactions between inflammation, the regulation of 5-HT metabolism, and different 5-HT signaling pathways in the gut. These will be outlined in the following sections.

#### 2.4.1. Regulation of 5-HT by TpH1 and SERT Affects Inflammation

Animal models show clear effects of TpH1 on inflammation. For example, in DSS-induced colitis, TpH1^−/−^ mice produce lower levels of the pro-inflammatory cytokines TNF-α, IL-1β, and IL-6 and have reduced macrophage infiltration when compared to TpH1^+/+^ mice [104]. Administration of 5-HTP, the 5-HT precursor, leads to an increased number of EC cells and higher amounts of 5-HT while the production of pro-inflammatory cytokines in the colon also increases [104]. In addition, TpH1^−/−^ mice with DNBS-induced colitis showed a reduced severity of the disease [104]. In other experiments, DCs from TpH1^−/−^ mice with DSS-induced colitis produced much lower amounts of IL-12p40, confirming a possible pro-inflammatory role of 5-HT on DCs [103]. Moreover, TpH1 deletion in mice reduced the impact of DCs on T cells resulting in lower production of IL-17 and IFN-γ [103].

SERT^−/−^ mice show significantly reduced health conditions and survival rates compared to littermate controls when suffering from colitis induced by 2,4,6-trinitrobenzene sulfonic acid (TNBS) [114]. When colitis in mice was induced via IL-10^−/−^, it was shown that the disease worsened in combination with SERT deletion, which was accompanied by increased levels of IL-6 and TNF-α mRNAs [115]. Expression of IL-10 is known to upregulate SERT production through the phosphatidylinositol 3 kinase pathway [116]. Thus, increased 5-HT availability seems to have a pro-inflammatory role during intestinal inflammation. In line with this, a recent study showed reduced *SLC6A4* (encoding SERT) expression in epithelium of patients with active CD and UC compared with healthy controls. Incubation of patient-derived colon epithelial cell lines with TNF-α also results in reduced SERT expression [117].

#### 2.4.2. Altered Abundances of EC Cells and 5-HT during Inflammation

Colonic endocrine cell types in patients with CD and UC demonstrated a significant increase in the cell number of 5-HT-immunoreactive cells in the colonic mucosa of these patients [118]. Similarly, a study demonstrated upregulation of EC cell numbers in mice with dinitrobenzene sulfonic acid (DNBS)-induced colitis, regulated by monocyte chemoattractant protein 1 [119]. In DSS-treated rats, an increase in epithelial EC cells in both the proximal and distal colon was also observed. This increase was accompanied by an increase in 5-HT content in the mucosal and submucosal tissue [73]. Consistently, it was shown that intestinal inflammation is responsible for a parallel two-fold increase in both the number of EC cells and the content of 5-HT, while SERT expression was reduced in the mucosa of inflamed colons in guinea pigs with TNBS-induced colitis [74].

In contrast, 5-HT levels in the inflamed mucosa of patients with CD and UC were observed to be markedly lower than in controls [120]. The authors of this study concluded that the decreased 5-HT content in inflamed tissues might result from a reduced ability of EC cells to produce 5-HT or from a reduction in the number of these cells [120]. In agreement, in another study, the number of EC cells, as well as the 5-HT content, was shown to be significantly reduced in the colon of patients with severe UC as well when compared to the non-severe UC patient group and healthy individuals [111]. It was concluded that there is a positive correlation between the 5-HT content of the colonic mucosa and the numbers of EC cells in the biopsy specimens from patients with UC. TpH1 and SERT expression was shown to be decreased in all UC samples [111].

Other studies demonstrate conflicting results. Mice with TNBS-induced colitis exhibited lower levels of SERT mRNA and lower intensity of SERT immunoreactivity in the mucosal tissue compared to control animals [75]. Researchers also observed higher amounts of 5-HT, which was attributed to the decrease in the expression of SERT while the number of EC cells remained the same [75]. In mice models infected with *Citrobacter rodentium*, ten days post-infection reduced numbers of EC cells, and reduced expression of SERT was observed [76]. However, overall increased production of 5-HT per individual EC cell was observed, which could be attributed either to the fact that EC cells produce more 5-HT during infection with *C. rodentium* or to the decreased reuptake of mucosal released 5-HT [76]. In line with this, SERT expression was shown to be significantly lower in inflamed mucosa of patients with UC than in healing mucosa [121]. In addition, in mice with DSS-induced colitis SERT expression was reduced in inflamed mucosa, which continued during the healing phase [121]. The same result was observed after the transfer of CD4^+^ T cells from mice models of spontaneous gastrointestinal inflammation (SAMP1/Yit) resembling Crohn’s disease to SCID mice that suffer from severe combined immunodeficiency affecting both B and T lymphocytes [121]. However, contrasting observations are described as well, showing increased mRNA levels of SERT in the ileum of patients with IBD [122].

Although the results described above are conflicting, it is clear that in IBD and animal models of IBD 5-HT synthesis, transport and signaling are often disturbed (Table 5 and Table 6).

#### 2.4.3. Neuronal Effects of 5-HT on Gut Motility, Enteric Neurogenesis and Maturation, and Wound Healing

Besides directly affecting immune cells, 5-HT can also influence gut motility and intestinal inflammation by acting on the neuronal system in the gut as a neurotransmitter. Neuronal 5-HT can induce gut motility and is involved in enteric neurogenesis and differentiation, which is modulated by microbiota [19,42,129]. It is suggested that the release of 5-HT stimulates HTR3 located on vagal sensory fibers resulting in the release of acetylcholine, subsequently leading to muscle contraction [49]. In addition, activation of HTR3, and the inositol 1,4,5-triphosphate pathway could cause the release of Ca^2+^, which induces contraction of colonic myocytes [49]. Dysregulation of the intestinal 5-HT metabolism was described in patients with both diarrhea and constipation-predominant irritable bowel syndrome (IBS) [130,131,132].

Patients with CD that are in remission and experiencing IBS-like symptoms also show upregulated levels of colonic mucosal TpH1, indicating that neuronal 5-HT may be a catalyst in the generation of the symptoms [133]. Indeed, TpH1 seems to play a role in gut motility as it was shown that TpH1^−/−^ mice exhibited no peristaltic reflexes and had altered fecal pellets [134]. However, another study comparing TpH1^−/−^ and TpH2^−/−^ mice showed that the intestinal transit time, the small bowel transit, and the colonic motility were decreased only in TpH2^−/−^ mice [135].

A further study showed that TpH2^−/−^ mice with DSS-induced colitis had increased severity of disease characterized by an increased secretion of the pro-inflammatory cytokines IL-1β, IL-6 and, TNF-α, and high mortality rates in comparison to the wild type [136]. Thus, besides affecting gut motility, the neuronal synthesis of 5-HT by TpH2 seems to play an anti-inflammatory role during intestinal inflammation [136]. Yet, it is not clear whether the deterioration of the disease was caused by altered gut motility or the absence of the 5-HT neuroprotective effect [136]. It was shown that during IBD, the number of enteric neurons is increased and that this could be associated with 5-HT-mediated neuroprotection and neurogenesis through the activation of neuronal HTR4 receptors [137].

Several studies implicated a positive role for 5-HT signaling in mucosal healing. Using TpH1^−/−^ and TpH2^−/−^ mice, it was shown that neuronal but not mucosal 5-HT was found to promote growth and proliferation of epithelial cells, in particular via HTR2A [138]. In accord, activation of HTR4 increased cell proliferation in intestinal crypts and promoted cell migration while reducing oxidative stress and apoptosis in Caco2 cells [127]. Similar effects on cell proliferation have been described in a keratinocyte HaCaT cell line, hepatic stellate cells, and pulmonary arterial smooth muscle cells [139,140,141]. These findings suggest that HTR activation could augment disease characterized by mucosal damage like IBD through stimulation of epithelial cell proliferation and hence mucosal healing. Yet, it should be noted that mucosal healing is a multifaceted process, with cell proliferation being only one aspect.

### 2.5. Targeting 5-HT Metabolism and Signaling in IBD

Because a role of 5-HT in intestinal inflammation seems likely, attempts were made to target 5-HT signaling to develop IBD therapeutics. In this section, the current efforts intending to ameliorate IBD by restoring 5-HT signaling through the administration of peripheral TpH1 inhibitors and selective serotonin reuptake inhibitors (SSRIs) will be described. Next, interventions targeting 5-HT signaling by 5-HTR will be outlined.

#### 2.5.1. Inhibition of 5-HT Synthesis by Blocking TpH1

The difference between TpH1^−/−^ mice TpH2^−/−^ mice in the severity of colitis due to reduced levels of enteric 5-HT and reduced neuronal 5-HT, respectively, indicates that 5-HT can have different effects depending on the location [104,136]. This makes intervening in these pathways challenging. Treatment strategies should target local effects rather than global inhibition. Since the development of drugs that specifically inactivate intestinal TpH1 has not yet been achieved, non-specific TpH1 inhibitors that cannot affect TpH2 expression in the central and enteric nervous system are currently in use (e.g., LP-920540, telotristat etiprate (LX1032; LX1606), parachlorylphenylalanine) [142]. A disruption in TpH2 expression needs to be avoided, as it would lead to abnormal behavior and complications in GI motility and neurogenesis [20,135].

Oral administration of two peripheral TpH1 inhibitors, LP-920540 and telotristat etiprate, in mice with TNBS-induced colitis, resulted in improvement of the disease state and reduced expression of inflammatory cytokines and chemokines in the intestine [143]. Moreover, 5-HT synthesis in the ENS, as well as GI motility, were not affected [143]. Similar results were observed for telotristat etiprate when administered in mice treated with DSS-induced colitis or infected with *Trichuris muris,* disease severity was delayed, and less histological damage, reduced myeloperoxidase activity, and decreased pro-inflammatory cytokine production was observed. Administration of another TpH1 inhibitor, parachlorylphenylalanine, in DSS-treated mice showed a decrease in disease severity and lower macroscopic and histologic scores in comparison to wild-types [104]. This health improvement in mice was accompanied by decreased macrophage infiltration, pro-inflammatory cytokine secretion, and lower serum levels of myeloperoxidase as well as C-reactive protein [104].

#### 2.5.2. SSRIs

SSRIs are a type of antidepressant that acts by blocking the SERT transporter and increasing the availability of 5-HT in the synaptic space [144]. Although typically prescribed to treat depression and anxiety, SSRIs also seem to exhibit anti-inflammatory properties and affect the gut microbiota composition [145]. For instance, fluoxetine allows higher expression of sporulation factors and membrane transporters in *Turicibacter sanguinis* by inhibiting the suppressive function on these processes by 5-HT, thereby reducing colonization in the gut [71].

Although SERT expression is reduced in IBD [111] and the observation that SERT^−/−^ rodents suffer from more severe chemically-induced colitis, SSRI administration in animal models seems to play a protective role in gut inflammation. More specifically, two compounds being fluoxetine and fluvoxamine, have been proven to have anti-inflammatory effects in IBD animal models. Pre-treating mice with fluoxetine before inducing colitis with DSS attenuates the severity of the disease as was shown by the colon length and reduced histological damage accompanied by suppression of the NF-κB pathway in intestinal epithelial cells and reduced neutrophil recruitment [146]. Fluoxetine also did not affect 5-HT release from EC cells. However, it should be noted that in this study, higher concentrations of the drug were administered than the usual dosage given to patients. In another study, fluoxetine also improved disease of IL-10^−/−^ mice and reduced TNF-α and IL-12p40 secretion [147]. Fluvoxamine ameliorated intestinal inflammation in mice with acetic acid-induced colitis by inhibiting the NF-κB pathway in intestinal epithelial cells [148].

It is hard to evaluate the efficacy of SSRI in patients with IBD due to the lack of randomized trials [145]. Interestingly, two case-control studies have shown that the administration of SSRIs can increase the risk of microscopic colitis [149,150].

#### 2.5.3. Targeting HTR Signaling

Due to the extended distribution of HTRs on gut and immune cells and the great impact they have on the onset of intestinal inflammation, many attempts have focused on modulating these receptors in an effort to ameliorate inflammation.

Blockade of HTR1A via the antagonist WAY100135 deteriorated TNBS-induced colitis in mice and impaired systemic neutrophil recruitment while the HTR1A agonist 8-OH-DPAT improved disease state, which is hypothesized to be achieved by a reduction in the 5-HT amount through activation of HTR1A on EC cells, indicating a negative feedback loop [97]. Conversely, blocking HTR2A with Ketanserin in TNSB-induced mice downregulated the expression of pro-inflammatory cytokines, neutrophil recruitment, and colonic apoptosis, and thereby improved animals health [97]. However, stimulation of HTR2A via the agonist (R)-DOI in mice that were characterized by TNF-α induced inflammation showed anti-inflammatory responses in the small intestine and in the aortic arch [123]. Specifically, genes correlated with the expression of pro-inflammatory cytokines, such as IL-6, and chemokines and cell adhesion proteins, such as *Vcam-1*, were upregulated [123].

Studies on HTR3 demonstrated that the antagonists’ tropisetron and granisetron had a positive effect on improving the health conditions of rats induced with acetic acid colitis [124,125]. Tropisetron reduced colonic damage, which was observed both macroscopically and microscopically. Tropisetron also reduced neutrophil infiltration and inflammatory cytokine production, including TNF-α, IL-1b, and IL-6 [124]. Similar results were observed by the use of granisetron [125]. Another HTR3 antagonist, ondansetron, restricted diarrhea in croton oil-induced colitis in murine models [126].

Mice induced with DSS- or TNBS-induced colitis receiving the HTR4 agonist tegaserod by enema exhibited improvement in disease severity in comparison to wild-type mice with inhibited HTR4 or HTR4^−/−^ mice, which was associated with upregulated proliferation and migration of colonic epithelial cells and decreased apoptosis caused by oxidative stress [127]. Gut motility was also found to be increased in the inflamed colons of mice and guinea pigs. On the contrary, all these effects conversed with rectal administration of HTR4 antagonist GR113808, which induced colitis in non-inflamed colons of wild-type mice [127].

In the case of HTR7, evidence from two studies exhibits differential results. In one study, the HTR7 was blocked in mice with DSS-induced colitis with the antagonist SB-269970. This had a positive effect on the health of mice, as histological damage was restricted and pro-inflammatory cytokine levels were reduced compared to control mice [128]. In line with this, it was also observed that the severity of colitis was reduced in HTR7^−/−^ mice in DSS- and DNBS-induced colitis models [128]. In contrast, in another study, it was observed that blockade of HTR7 in DSS-induced colitis or in IL-10^−/−^ mice with the antagonist SB-269970 resulted in increased inflammation [112]. The same result was observed in HTR7^−/−^ mice post-DSS-induced colitis, in which IL-1β was upregulated. Conversely, HTR7 stimulation via the agonist 5-carboxamidotryptamine maleate had an anti-inflammatory effect [112]. The authors support that discrepancies between the two studies are due to differential experimental design and, specifically, differences in drug administration (orally versus intraperitoneal injection), drug concentrations, and housing conditions (conventional conditions versus specific pathogen-free housing) [112]. Table 6 describes the overall impact of HTR modulation on gut inflammation in animal IBD models.

## 3. Discussion

This review describes the interactions between microbial and host 5-HT metabolism, affecting immune responses, gut motility, and wound healing. An overview of these interactions is provided in Figure 2.

Gut microbiota competes with the host for the available tryptophan, and the products of both host and microbial tryptophan metabolism activate immune responses that promote gut homeostasis. Microbes contribute to 5-HT synthesis either by direct production of 5-HT, as was shown at least in vitro, or through the secretion of metabolites, like SCFAs and secondary bile acids that can stimulate EC cells to produce 5-HT [40]. This suggests that gut microbes can reduce the host’s tryptophan availability but increase 5-HT availability. Host-derived 5-HT can influence gut microbiota composition by promoting the colonization of specific species against others, and 5-HT can worsen intestinal inflammation through the suppression of beneficial microbes that enhance gut homeostasis both directly and indirectly via the inhibition of antimicrobial peptides. However, other studies show that host-derived 5-HT induces the colonization by microbial SCFAs producers, which are overall associated with health. Thus, although host 5-HT likely affects the microbial composition in the gastrointestinal tract, more research needs to be done in order to evaluate whether these effects are beneficial or harmful.

5-HT likely plays a major role in intestinal inflammation by either interacting with local immune cells via specific HTRs or by stimulating immune responses when released from activated platelets. Overall, 5-HT drives intestinal immune cells to activate both pro- and anti-inflammatory pathways depending on the receptor subtype expressed on the immune cells.

According to the studies on 5-HT signaling in IBD, the number of EC cells can be either upregulated or downregulated both in animal and human models. 5-HT content seems to be increased in animal IBD models and decreased in human IBD models or patients. These differences could be attributed to the different models of colitis that are used in the experiments. The severity and location of the disease may also play a role in these changes, as heavily inflamed tissues are characterized by fewer EC cells and lower 5-HT levels than less severe cases. SERT expression is reduced in the vast majority of the models examined, which means that less 5-HT is degraded into enterocytes, and 5-HT levels are increased. Further evidence that 5-HT may have a pro-inflammatory role comes from the observation that TpH1^−/−^ and SERT^−/−^ animals exhibit reduced inflammation in various IBD models. On the other hand, neuronal 5-HT seems to have a more anti-inflammatory role, possibly by its effect on gut motility, as evidenced by increased severity of inflammation in TpH2^−/−^ animal models.

Selective peripheral TpH1 inhibitors seem to be promising compounds for the treatment of IBD since studies show that their administration in animal IBD models ameliorated intestinal inflammation. They are also likely to be safe since they do not gain access to central and enteric serotonergic neurons and do therefore not affect behavior, gut motility, and neuroprotection. SSRIs, such as fluoxetine and fluvoxamine, seem to have anti-inflammatory properties in animal IBD models. However, their impact on patients with IBD is still doubtful, and larger trials based on measurements of anti-inflammatory markers and histology need to be performed. Moreover, there are indications that SSRIs can affect the gut microbiota as alterations in abundances were observed when treating with SSRIs in animal models for depression [71,151,152]. More research needs to be done in the future in order to examine to what extent these implications can be hazardous or beneficial for the host. Interpretation of results is more complicated for HTR agonists or antagonists tested in animal models, and this is probably due to the high diversity of the HTRs and the differences in the experimental design. Nevertheless, from the data collected, it can be assumed that HTR1A and HTR4 stimulation have anti-inflammatory effects while HTR3 activation increases intestinal inflammation. The impact of HTR2A and HTR7 on intestinal inflammation is still controversial.

It should be noted that this review describes the role of 5-HT signaling in the intestine with a focus on IBD. Functional gastrointestinal disorders such as IBS are now defined as “Disorders of gut-brain interaction” [153]. This definition indicates the complex etiology [154] of these disorders, both local intestinal and central nervous system processes are involved. For a discussion of the importance of 5-HT signaling in IBS and the use of drugs that interfere with 5-HT signaling in the treatment of IBS, we, therefore, refer to some excellent recent reviews [155,156,157].

## 4. Conclusions

This review shows the importance of 5-HT in the intestine but also presents us with a complex picture. Microbial, epithelial, and immune components of the intestine are responsive to 5-HT signaling but also use 5-HT in signaling routes that affect intestinal homeostasis. Further exploring the complex intestinal 5-HT signaling network might lead to novel insights that could benefit patients suffering from inflammatory bowel disease.

## Figures and Tables

**Figure 1 ijms-22-09487-f001:**
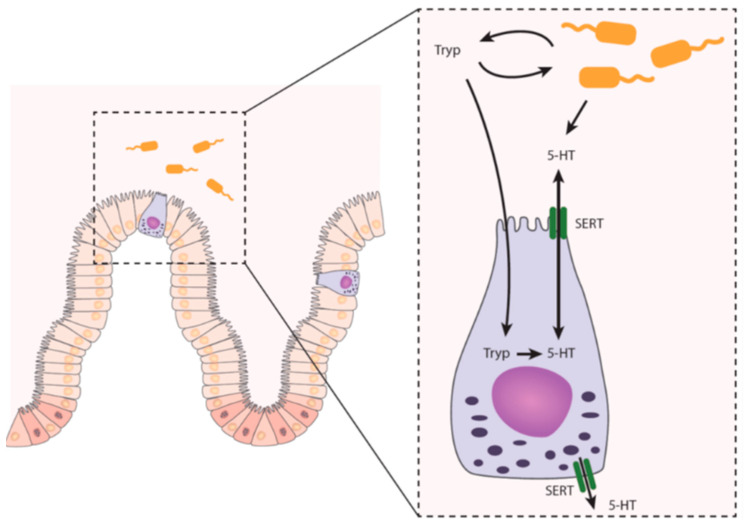
Intestinal 5-HT metabolism. Enteroendocrine cells in the gut epithelium synthesize 5-HT from tryptophan. This process is directly (via the availability of tryptophan) and indirectly (promotion of synthesis by microbial metabolites) regulated by the gut microbiota. The release of 5-HT is mediated by SERT. In addition, the gut microbiota might also be able to produce 5-HT. 5-HT: 5-hydroxytryptamine; SERT: serotonin reuptake receptor; Tryp: tryptophan.

**Figure 2 ijms-22-09487-f002:**
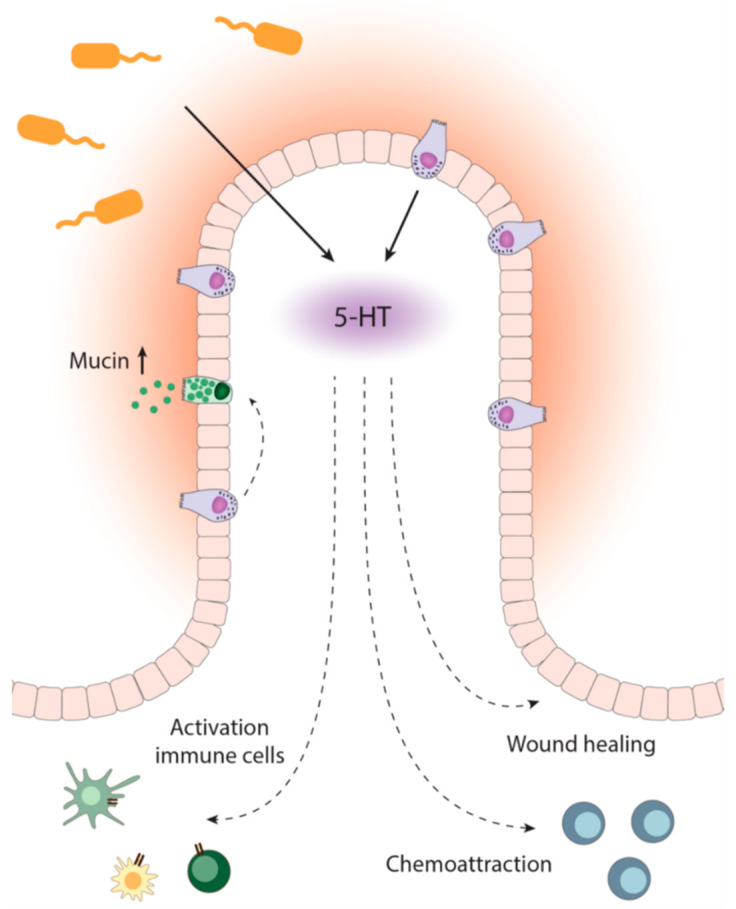
Overview of 5-HT action in the intestine. 5-HT is produced by microbes and EC cells. Subsequently, immune cells are activated, and processes such as chemoattraction and wound healing are activated—furthermore, mucus production by epithelial goblet cells increases. 5-HT: 5-hydroxytryptamine; EC: enterochromaffin.

**Table 1 ijms-22-09487-t001:** Expression of TpH1 and SERT by immune and epithelial cells.

Cell Type	TpH1	SERT
B cells [15]		+
DCs [14]		+
Mast cells [12]	+	+
Macrophages [10,13]	+	+
Monocytes [10,11]	+	+
T cells [13,16]	+	+
Platelets [21,22]		+
Enterocytes [23]		+
EC cells [19,24,25]	+	+

DCs: dendritic cells; EC: enterochromaffin; TpH1: tryptophan hydroxylase 1; SERT: serotonin reuptake receptor.

**Table 2 ijms-22-09487-t002:** Bacterial species capable of 5-HT synthesis.

Bacterial Strains	Present in Human Gut	Phylum	Family
*Lactococcus lactis* subsp. *cremoris* (MG 1363) [51]	Used as probiotic	Firmicutes	Streptococcaceae
*Lactococcus lactis subsp. lactis* (IL1403) [51]	Used as probiotic	Firmicutes	Streptococcaceae
*Lactobacillus plantarum* (FI8595) [51]	+	Firmicutes	Lactobacillaceae
*Streptococcus thermophilus* (NCFB2392) [51]	Used as probiotic	Firmicutes	Streptococcaceae
*Escherichia coli* K-12 [52]	+	Proteobacteria	Enterobacteriaceae
*Morganella morganii* (NCIMB, 10466) [53]	+	Proteobacteria	Morganellaceae
*Klebsiella pneumoniae* (NCIMB, 673) [50,53]	+	Proteobacteria	Enterobacteriaceae
*Hafnia alvei* (NCIMB, 11999) [53]	+	Proteobacteria	Hafniaceae
*Corynebacterium* sp. [50]	+	Actinobacteria	Corynebacteriaceae
*Aeromonas* [50]	+	Proteobacteria	Aeromonadaceae
*Citrobacter* [50]	+	Proteobacteria	Enterobacteriaceae
*Enterobacteria aglomerans* [50]	Pathological conditions	Proteobacteria	Erwiniaceae
*Shigella* [50]	+	Proteobacteria	Enterobacteriaceae
*Achromobacter xylosoxidans* [50]	+	Proteobacteria	Alcaligenaceae
*Chromobacterium* [50]	Pathological conditions	Proteobacteria	Neisseriaceae
*Acinetobacter* [50]	+	Proteobacteria	Moraxellaceae
*Listeria monocytogenes* [50]	+	Firmicutes	Listeriaceae
*Staphylococcus aureus* [50]	+	Firmicutes	Staphylococcaceae

**Table 3 ijms-22-09487-t003:** Expression of HTRs by immune and epithelial cells.

Cell Type	Receptor Subtype (HTR-)
B cells [82]	1A, 2, 3, 7
Immature dendritic cells (DCs) [83]	1B, 1E, 2A, 2B, 3
Mature dendritic cells (DCs) [83]	2A, 3, 4, 7
Eosinophils [84]	1A, 1B, 1E, 2A, 2B, 6
Mast cells [85]	1A, 1B, 1E, 2A, 2B, 2C, 3, 4, 7
Macrophages [10,82,86,87]	1A, 2A, 2B, 2C, 3, 4, 7
Monocytes [88]	1E, 2A, 3, 4, 7
Neutrophils [82]	7
NK [82]	1A, 2A, 2B, 2C
T cells [89,90,91,92]	1A, 1B, 2A, 7
Platelets [93]	2A, 3
Paneth cells [94]	2A
Enterocytes [94,95]	2A, 4
Goblet cells [95,96]	4
EC cells [95,97]	1A, 4

DC: dendritic cell; EC: enterochromaffin; NK: natural killer.

**Table 4 ijms-22-09487-t004:** Effects of HTR stimulation on immune and epithelial cells.

Cell Type	Sample Source and 5-HTR Stimulation	Effect
Basophils	Mice (IL-3 co-stimulation) [99]	IL-4 ↓ IL-6 ↓ Histamine secretion ↓
	Human blood (IL-3 co-stimulation) [99]	IL-4 ↓ IL-13 ↓
B cells	Murine and rat spleen cells (HTR1A) [100]	Proliferation ↑
Immature DCs	Human monocyte-derived DCs (HTR1 and HTR2) [101]	Migration ↑
Mature DCs	Mice (HTR7) [102]	Migration ↑
	Human monocyte-derived DCs (HTR3, HTR4, and HTR7) [101]	IL-6 ↑
	Human DCs (HTR4 and HTR7) [83]	IL-1β ↑ IL-8 ↑
	Mice with DSS-induced colitis [103]	IL-12p40 ↑
Eosinophils	Human eosinophils from patients with asthma and/or rhinitis (HTR2A) [84]	Recruitment of eosinophils Epithelial adhesion ↑ (VCAM-1)
Mast cells	Murine bone marrow-derived mast cells and human CD34^+^ mast cells [12]	Epithelial adhesion ↑ (fibronectin) Migration ↑
Macrophages	Peritoneal cavity of TpH1^−/−^, and DSS-induced colitis in mice with or without LPS stimulation (HTR1A) [104]	IL-1β ↑ IL-6 ↑ TNF-α ↑
	Murine peritoneal macrophages (HTR1A) [86]	Phagocytosis ↑
	Human monocyte-derived macrophages (HTR2B and HTR7) [10]	Anti-inflammatory polarization
Monocytes	Rats with TNBS-induced colitis [105] Human peripheral blood mononuclear cells (HTR3, HTR4, and HTR7) [88] Human peripheral blood mononuclear cells (HTR7) [88]	Adhesion to colonic epithelial cells ↑ IL-6 ↑ IL-8 ↑ MCP-1 ↑ IL-1β ↑ IL-6 ↑ IL-8 ↑ TNF-α ↓ IL-12p40 ↑
Neutrophils	TpH1^−/−^ mice and mice with acute peritonitis, lung inflammation, and aseptic skin wounds [106]	Recruitment of neutrophils
NK cells	Human peripheral blood mononuclear cells (HTR1A) [107]	Cytotoxicity ↑ Interaction with monocytes ↓
T cells	Murine spleen cells (HTR7) [89] Human peripheral blood mononuclear cells (HTR1B) [90] Mice (HTR2A) [91]	Proliferation of naive T cells ↑ Proliferation of Th cells ↑ IFN-γ ↑ and IL-2 ↑ in Th1 and CTL cell line
Platelets	TpH1^−/−^ mice (HTR2A, HTR3) [93,98]	Ca^2+^ release ↑ Exocytosis of α-granules ↑ Stabilization of platelet activation ↑
Paneth cells	HTR2A^−/−^ murine cells (HTR2A) [94]	Paneth cells density ↓
Enterocytes	HTR2A^−/−^ murine cells (HTR2A)^96^ Murine cells (HTR4) [108]	Enterocytes size ↓ Colonic transit
Goblet cells	Murine cells (HTR4) [108] Human intestinal enteroids (HTR4) [96]	Colonic transit Goblet cell degranulation TFF3 release ↑ Actin cytoskeleton rearrangement and epithelial repair
EC cells	Murine cells (HTR4) [109] TNBS-induced colitis in mice (HTR1A) [97]	Colonic transit 5-HT content ↓

5-HT: 5-hydroxytryptamine; DC: dendritic cell; DSS: dextran sulfate sodium; EC: enterochromaffin; HTR: 5-hydroxytryptamine receptor; IFN-γ; interferon γ; IL: interleukin; LPS: lipopolysaccharide; MPS-1: monocyte chemoattractant protein 1; NK: natural killer; TFF: trefoil factor; TNBS: 2,4,6-trinitrobenzene sulfonic acid; TNF-α; tumor necrosis factor α; TpH: tryptophane; VCAM-1: vascular cell adhesion molecule 1.

**Table 5 ijms-22-09487-t005:** Alterations in intestinal 5-HT signaling in animal IBD models and in patient-derived material.

Organism	Study	Experimental Design	Effect
Mouse	Khan et al. (2006) [119]	DNBS-induced colitis in mice	EC cell numbers ↑
	Oshima et al., (1999) [73]	DSS-induced colitis in rats	EC cell numbers ↑ 5-HT content ↑
	Linden et al. (2003) [74]	TNBS-induced colitis in guinea pigs	EC cell numbers ↑ 5-HT content ↑ SERT expression ↓
	Linden et al. (2005) [75]	TNBS-induced colitis in mice	Unchanged EC cell numbers 5-HT content ↑ SERT expression ↓
	O’Hara et al. (2005) [76]	Mice infected with *Citrobacter rodentium*	EC cell numbers ↓ 5-HT content ↑ SERT expression ↓
	Tada et al. (2016) [121]	DSS-induced colitis/Transfer of CD4+ T cells in mice	SERT expression ↓
Human	El-Salhy et al. (2020) [118]	Patients with CD and UC	EC cell numbers ↑
	Magro et al. (2002) [120]	Patients with CD and UC	5-HT content ↓
	Coates et al. (2004) [111]	Patients with severe UC	EC cell numbers ↓ 5-HT content ↓ SERT expression ↓ TpH1 expression ↓
	Tada et al. (2016) [121]	Inflamed mucosa of patients with UC	SERT expression ↓
	Wojtal et al. (2009) [122]	Patients with CD and UC	SERT expression ↑

5-HT: 5-hydroxytryptamine; CD: Crohn’s disease; DNBS: dinitrobenzene sulfonic acid; DSS: dextran sulfate sodium; EC: enterochromaffin; SERT: serotonin reuptake receptor; TNBS: 2,4,6-trinitrobenzene sulfonic acid; TpH1: tryptophan hydroxylase 1; UC: ulcerative colitis.

**Table 6 ijms-22-09487-t006:** Impact of 5-HT signaling on gut inflammation in animal IBD models.

HTR	Agonist/ Antagonist	Compound	Experimental Design	Impact on Inflammation
HTR1A	Antagonist	WAY100135	Mice with TNBS-induced colitis [97]	↑
	Agonist	8-OH-DPAT	Mice with TNBS-induced colitis [97]	↓
HTR2A	Antagonist	Ketanserin	Mice with TNBS-induced colitis [97]	↓
	Antagonist	M100907	Mice with TNF-α -induced inflammation [123]	↓
	Agonist	(R)-DOI	Mice with TNF-α-induced inflammation [123]	↓
HTR3	Antagonist	Tropisetron	Rats with acetic acid-induced colitis [124]	↓
	Antagonist	Granisetron	Rats with acetic acid-induced colitis [125]	↓
	Antagonist	Ondansetron	Murine models of croton oil-induced colitis [126]	↓
HTR4	Agonist	Tegaserod	Mice with TNBS and DSS-induced colitis [127]	↓
	Antagonist	GR113808	Wild-type mice [127]	↑
HTR7	Antagonist	SB-269970	Mice with DSS-induced colitis [128]	↓
	Antagonist	SB-269970	Mice with DSS and IL-10-induced colitis [112]	↑
	Agonist	5-carboxamidotryptamine maleate	Mice with DSS-induced colitis [112]	↓

DSS: dextran sulfate sodium; HTR: 5-hydroxytryptamine receptor; TNBS: 2,4,6-trinitrobenzene sulfonic acid; TNF: tumor necrosis factor.

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
