# Peer review of "The Multifaceted Role of Serotonin in Intestinal Homeostasis"

_ijms, 2021, doi:10.3390/ijms22179487_

Round 1
Reviewer 1 Report
The authors presented a very detailed review of the important monoamine-serotonin regulatory role at the gut microbiome-gut epithelium interface. The review presents a detailed analysis of experimental animal models, as well details the role of endogenous and exogenous monoamine-serotonin in the modulation of inflammatory processes of various origins. This review will undoubtedly be of interest to both specialists conducting scientific research in this area and clinical practitioners.
I only have one comment for this review:
The authors do not cite previous reviews on this research topic
( Coates MD, Tekin I, Vrana KE, Mawe GM. Review article: the many potential roles of intestinal serotonin (5-hydroxytryptamine, 5-HT) signalling in inflammatory bowel disease. Aliment Pharmacol Ther. 2017 Sep;46(6):569-580. doi: 10.1111/apt.14226. Epub 2017 Jul 24. PMID: 28737264.
Shajib MS, Baranov A, Khan WI. Diverse Effects of Gut-Derived Serotonin in Intestinal Inflammation. ACS Chem Neurosci. 2017 May 17;8(5):920-931. doi: 10.1021/acschemneuro.6b00414. Epub 2017 Mar 27. PMID: 28288510. )
although they are familiar with the publications of these authors and cite their research articles.
From my point of view, it would be useful to provide links to these reviews at least in the "introduction", as well as clearly indicate those aspects that distinguish the presented analytical review from the previous ones published earlier. Moreover, in this review, the "introduction" section is practically absent.
Reviewer 2 Report
The review by Koopman et al. is a detailed discussion of the role of serotonin in the many different cell types in the intestine. I found the work to be interesting and complete, with many important observations. I find no major issues and only minor grammatical mistakes. I have only a few comments.
1) There have been a few reviews of the role of serotonin in the intestine (for example, Terry and Margolis, 2017; Banskota et al, 2019). It would be important for this review to discriminate itself from the others in some way. What new information does this review bring that the others do not have?
2) I find the role of serotonin in intestinal health fascinating, with many possibilities to treat IBS (and perhaps other ailments). With the many patient years of SSRI use, I am surprised that there is not more known about the effect of SSRI use to treat mental health disorders on intestinal health. IF there is more known, it is important that more information about that be included in this review.
3) Given the extensive review of the effect of 5-HT on the various cells of the intestine, could the authors provide some comments about supplementation of the diet wth tryptophan or supplementation of probiotics with bacteria that produce more serotonin as a treatment for IBS?
Reviewer 3 Report
- Why paragraph 2.1.5-. is before 2.1.1?
- “Inflammatory bowel disease (IBD),”
Define the abbreviation the first time you use it
- “Inflammatory bowel disease (IBD), is characterized by chronic intestinal inflammation.”
Add a citation (for example, “Actis GC et al. History of Inflammatory Bowel Diseases. J Clin Med. 2019 Nov 14;8(11):1970. doi: 10.3390/jcm8111970. PMID: 31739460; PMCID: PMC6912289.”)
- “CD and UC patients” “CD patients”
Use “patients with CD and UC” etc.
- If 5-HT induces inflammation, what about the use of tryptophan (5-HT precursor) or SSRI (the increase the level of 5-HT) for example in irritable bowel syndrome?
- “histamine, IL-4 and IL-6”
Use oxford comma in the whole text
- In the introduction explain what your review adds to the reviews already available in the literature
- In the conclusion, suggest for clinicians if 5-HT, at the end, has a pro-inflammatory or anti-inflammatory effect
Author Response
Please se attachment
